# Anti-Proliferative Potential of Quercetin Loaded Polymeric Mixed Micelles on Rat C6 and Human U87MG Glioma Cells

**DOI:** 10.3390/pharmaceutics14081643

**Published:** 2022-08-06

**Authors:** Sathishbabu Paranthaman, Chinnappa A. Uthaiah, Riyaz Ali M. Osmani, Umme Hani, Mohammed Ghazwani, Ali H. Alamri, Adel Al Fatease, SubbaRao V. Madhunapantula, Devegowda Vishkante Gowda

**Affiliations:** 1Department of Pharmaceutics, JSS College of Pharmacy, JSS Academy of Higher Education and Research, Mysuru 570015, Karnataka, India; 2Centre of Excellence in Molecular Biology and Regenerative Medicine (CEMR) Laboratory, (a DST-FIST Sponsored Centre), Department of Biochemistry (a DST-FIST Sponsored Department), JSS Medical College, JSS Academy of Higher Education & Research (JSS AHER), Mysuru 570015, Karnataka, India; 3Department of Pharmaceutics, College of Pharmacy, King Khalid University, Guraiger, Abha 61421, Saudi Arabia; 4Cancer Research Unit, King Khalid University, Guraiger, Abha 61421, Saudi Arabia

**Keywords:** quercetin, anticancer activity, glioma, polymeric mixed micelles, Box–Behnken Design

## Abstract

Quercetin (Qu) is a natural flavonoid present in many commonly consumed food items and is also identified as a potential anticancer agent. The present study evaluates the Qu-loaded polymeric mixed micelles (Qu-PMMs) against C6 and U87MG glioma cell lines. The Box–Behnken Design (BBD) was employed to study the influence of independent variables such as Soluplus, Vitamin-E polyethyleneglycol-1000 succinate (E-TPGS), and poloxamer 407 concentrations on dependent variables including particle size (PS), polydispersity index (PDI), and percentage entrapment efficiency (%EE) of the prepared Qu-PMMs. The Qu-PMMs were further characterized by Fourier Transform Infrared Spectroscopy (FTIR), X-ray Diffraction (XRD), Scanning Electron Microscope (SEM), and were assessed for in vitro drug release, effect on cell viability, migration, cellular uptake, and apoptosis assays. The PS, PDI, and % EE of the optimized PMMs were 107.16 ± 1.06 nm, 0.236 ± 0.053, and 77.46 ± 1.94%, respectively. The FTIR and XRD revealed that the Qu was completely entrapped inside the PMMs. The SEM analysis confirmed the spherical shape of micelles. The in vitro cell viability study showed that the Qu-PMMs had 1.7 times higher cytotoxicity against C6 and U87MG cells than Qu pure drug (Qu-PD). Furthermore, Qu-PMMs demonstrated superior cellular uptake, inhibited migration, and induced apoptosis when tested against C6 and U87MG cells than pure Qu. Thus, the polymeric mixed micelle (PMMs) enhanced the therapeutic effect of Qu and can be considered an effective therapeutic strategy to treat Glioma.

## 1. Introduction

Gliomas are the most common and lethal primary solid tumor of the central nervous system (CNS). The lower survival rate (˂5 years) causes it to be more complicated, even with modern treatments such as radiotherapy, chemotherapy, and surgical resection [1,2,3]. However, these therapies were considered inefficient in managing gliomas not only because of the low survival rate (<15 months) but also due to the development of resistance and various toxic side effects [4,5].

Quercetin (Qu) is the most abundant polyphenolic bioflavonoid belonging to the flavones subgroup of flavonoid compounds [6]. Qu is regarded as a disease-preventing nutritional supplement due to potential health benefits. Several studies have reported the potential effect of Qu in treating various types of cancers, cardiovascular, and neurodegenerative diseases. In addition, Qu is a potent antibacterial and anti-inflammatory agent in addition to its free radical scavenging activity [7]. The Qu has been reported to act by several mechanisms, including cell growth inhibition, the induction of cell death (apoptosis) by modulating various signaling pathways in the brain, breast, colon, lung, ovarian, pancreatic, and rectal cancer cells [8,9,10]. Despite these beneficial health effects, the clinical application of Qu is limited due to low bioavailability, extensive metabolism, and rapid elimination from the body [11].

Various approaches have improved the Qu solubility, bioavailability, and cellular uptake. The delivery system possibly improves its limitations with promising features such as ease of preparation, stability, high loading, and encapsulation efficiency [12,13,14,15]. The polymer-based preparations are one such delivery system that fulfills the aforementioned features and is thus advantageous over the other conventional systems such as solutions, suspensions, self-emulsifying systems, and emulsion [10].

The polymeric micelles (PMs) are composed of amphiphilic copolymers that self-assemble into nanostructure (20 to 200 nm), and these PMs are among the most studied nanocarriers in the field of diagnosis and pharmacotherapy. This thermodynamically driven process occurs above a copolymer-determined concentration, commonly known as critical micellar concentration (CMC) [16,17]. PMs are formed by an inner hydrophobic core, in which poorly-water soluble drugs can be entrapped, and by an outer hydrophilic shell that protects the encapsulated drug from the external medium. However, the polymeric mixed micelles (PMMs) are more advantageous over other PMs in terms of drug loading, accurate particle size control, thermodynamic, and kinetic stability [18]. Recently, Qu-PMMs were developed using PLGA (poly (lactic-co-glycolic acid)) and polyvinyl alcohol by the single emulsion-solvent evaporation (SESE) method, wherein the cell proliferation and oxidation were reduced by enhancing the cellular uptake compared to pure Qu [10].

Soluplus (polyvinyl caprolactam (PCL)-polyvinyl acetate (PVA)-polyethyleneglycol (PEG)) is a biocompatible copolymer consisting of the hydrophilic part (PEG) as a backbone and PCL-PVA as a lipophilic sidechain structure with a critical micellar concentration (CMC)value of 7.6 mg L^−1^ in water. E-TPGS (vitamin E succinate with PEG) is an anionic conjugated copolymer of D-α-tocopheryl PEG 1000 succinate (TPGS) and vitamin E succinate. It improves the solubility and cellular uptake and prolongs the systemic residence of the drug. Notably, it also serves as a potent efflux blocker for the drugs from cells by inhibiting P-gp [19]. E-TPGS has also been used as a prominent nanocarrier for curcumin to improve its anticancer activity on human colon cancer cells [20]. A previous study revealed that the Soluplus and E-TPGS polymeric nanoparticles improved in vitro cytotoxicity and in vivo biodistribution of paclitaxel and resveratrol [21]. Poloxamer 407 (Kolliphor^®^ P 407) is a block copolymer that contains ethylene oxide and propylene oxide with a low CMC value of 34.2 mg L^−1^ at 37 °C. The nanoparticles using Soluplus and poloxamer 407 have been reported to enhance the bioavailability of Qu [22].

Thus, the present study focused on developing and optimizing the Qu-PMMs using Soluplus, E-TPGS, and Poloxamer 407 for the enhanced anti-cancer activity of Qu against Glioma cells (Figure 1). The optimized Qu-PMMs were characterized by Fourier Transform Infrared Spectroscopy (FTIR), X-ray Diffraction (XRD), Scanning Electron Microscope (SEM), and further evaluated for in vitro drug release, cellular uptake, cell viability, migration, and apoptosis.

## 2. Materials and Methods

### 2.1. Materials

Quercetin (Qu) was procured from Sisco Research Laboratories Pvt. Ltd., India. Soluplus and poloxamer 407 were obtained as a gift sample from the BSFA (British Scientific Fiction Association Lit.). E-TPGS was gifted from Shilpa medicare Ltd., Bangalore, India. Dulbecco’s Modified Eagle’s medium (DMEM), Trypsin EDTA, Dulbecco’s phosphate buffer 7.4 pH, Penicillin–Streptomycin, Fetal bovine serum (FBS), and glutamate were purchased from Thermo Fisher Scientific (Pittsburgh, PA, USA). Cell culture plasticware was obtained from Techno Plastic Products AG (TPP, Trasadingen, Switzerland). Dimethyl Sulfoxide (DMSO) molecular grade and 3-(4, 5-Dimethylthiazol-2-yl)-2, 5-diphenyl tetrazolium bromide) (MTT) were purchased from Sigma-Aldrich, USA. The water used in the experiment was RO purified, and all other solvents and chemicals used were of analytical grade.

### 2.2. High-Performance Liquid Chromatography (HPLC) Method Development for Qu

Qu was estimated using the previously established HPLC method with slight modifications [23]. The Shimadzu HPLC system (LC20C, *I* prominence series) consisting of an auto-sampler system and quaternary pump equipped with a UV-Visible detector was used for the analysis. Qu was eluted effectively by using a C18 column (Shimadzu, 5 μm particle size ODS, 150 mm × 4.6 mm) set at the temperature of 30 ± 2 °C, and the injection volume was 20 μL. The mobile phase constituted of methanol and 0.1% orthophosphoric acid in the ratio of 98.5:1.5. The flow rate was adjusted to 1.0 mL/min and detected at 361 nm. Seven concentrations of Qu from 400 to 2800 ng/mL were analyzed to evaluate the linearity of the method (*n* = 3).

### 2.3. Glioma Cell Line

Rat glioma cells (C6 ATCC^®^ CCL-107) and human glioma cells (U87MG ATCC^®^ HTB-14) were procured from American Type Culture Collection (ATCC, Rockville, MD, USA), were conserved employing DMEM media, augmented with 10% *v/v* FBS and 1% *v/v* Penicillin–Streptomycin and glutaMAXTM solution at 37 ± 0.5 °C with 5% carbon dioxide (CO_2_) and 95% relative humidity in a carbon dioxide incubator [24,25].

### 2.4. Preparation of Qu Loaded Polymeric Mixed Micelles (PMMs)

In the present study, a single emulsion solvent evaporation method was employed to develop Qu-PMMs [10]. Initially, the aqueous phase was prepared in a separate beaker by dissolving 0.1 to 0.3% (*w/v*) of poloxamer 407 in 10 mL of water. Qu (10 mg mL^−1^), Soluplus (10 to 20 mg mL^−1^), and E-TPGS (0.1 to 1.0 mg mL^−1^) were weighed, and methanol (5 mL) was added for solubilization and stirred on a magnetic stirrer at 300 RPM for 10 min. The oil phase was gradually transferred using a syringe to an aqueous phase subjected to a magnetic stirrer at 1000 rpm overnight (constant for all the formulations).

#### Screening and Optimization of Qu-PMMs by Box-Behnken Design (BBD)

Three-factor three-level BBD was employed to optimize the PMMs. The influence of three independent factors, including the various concentrations of Soluplus, E-TPGS, and poloxamer 407 on response variables, including the Particle size (PS), polydispersibility index (PDI), and percentage entrapment efficiency (% EE) of Qu-PMMs, were studied. The variables and levels of the BBD model are tabulated in Table 1. A total of 17 runs was defined using design Expert version 13.0, state Ease Inc., Minneapolis, MN, USA). The experimental runs were carried out in triplicates, and results were expressed as Mean ± Standard Deviation (SD).

The optimized independent polymer ratio was utilized to prepare the blank PMMs, and the physical appearance of prepared blank PMMs, Qu-PMMs, and schematic diagram of Qu-PMMs were shown in Figure 2I(A,B),II, respectively. The optimized blank and Qu-PMMs were lyophilized at −80 °C (without cryoprotectant), and the lyophilized powders were utilized for further in vitro evaluations.

### 2.5. Determination of Particle Size (PS), Poly Dispersibility Index (PDI), and Zeta Potential

The Blank PMMs and Qu-PMMs were evaluated for the PS, PDI, and zeta potential using a PS analyzer (Malvern Instruments, Worcestershire, UK, model ZS 90), which works based on dynamic light scattering. The PS and PDI of the formulations were measured at 25 °C. The samples were diluted using milli Q water (1:100) before measuring the PS and PDI.

### 2.6. Determination of Percentage Entrapment Efficiency (% EE)

The % EE of the Qu-PMMs formulations was measured using the centrifugation technique, wherein 1 mL of Qu-PMM was centrifuged at 15,000 RPM for 30 min using a high-speed centrifuge. The 10 µL of supernatant was diluted with methanol, and Qu was estimated by the HPLC method [21]. The % EE was calculated by using the following Equation (1):% EE = Total Qu − Qu in supernatant / Total Qu × 100(1)

### 2.7. Scanning Electron Microscopy (SEM)

The morphological characteristics of the Qu-PMMs were determined using a scanning electron microscope (Apreo S Themo Fisher Scientific Inc., Waltham, MA, USA). Qu-PMMs solution was affixed on gold-coated copper stubs and analyzed using an accelerating voltage of 20 kV [26].

### 2.8. Fourier Transform Infrared (FT-IR) Spectroscopy

The FT-IR (Shimadzu-8400S) analysis of Qu-PMMs was performed by employing the potassium bromide (KBr) pellet press technique. The pellets with pure Qu, Blank PMMs, and Qu-PMMs at a 1:10 ratio. The spectrum evaluations were performed within the range of 4000 cm^−1^ to 400 cm^−1^ [26].

### 2.9. Powder X-ray Diffraction (P-XRD) Analysis

P-XRD (PROTO) analysis for pure Qu, Blank PMMs, and Qu-PMMs was performed to study the drug’s physicochemical and structural changes in the PMMs. The diffraction pattern was studied using copper as a radiation source, with the scanning range and scan rate being 5° to 60° (2θ°) 0.6 s at 2θ and a step size of 0.0200° at 2θ, respectively [27].

### 2.10. In Vitro Drug Release Studies

The in vitro drug release study of Qu pure drug and Qu-PMMs was studied using PBS 7.4 pH with 1% tween 20 as a release medium by a dialysis method adopted from Lv et al., with minor modifications [28]. Pure drug Qu and Qu-PMMs were added to the previously soaked dialysis tube of Molecular Weight Cut off (MWCO) 12kD (Himedia Laboratories Pvt, Ltd., Mumbai, India.) and sealed at both ends. The tubes were then suspended in the beaker comprising 75 mL of the release media and placed on a stirrer at 100 rpm and 37 °C ± 2 °C temperature. The samples were withdrawn at frequent intervals (0.25, 0.5, 1, 2, 3, 4, 8, 12, 24, 36, 48, and 72 h) and replenished with the same quantity of release media to retain the sink condition. The obtained samples were then diluted with the appropriate mobile phase and quantified for Qu release from PMMs using the HPLC technique (LC-2030C, Shimadzu, Japan).

### 2.11. Stability Studies of Optimized PMMs

The optimized Qu-PMMs formulation was stored at 4 °C and 25 °C for 2 months for stability analysis, and the samples were analyzed for PS, PDI, and %EE at periodic intervals (0, 7, 15, 30, and 60 days) [29].

### 2.12. In Vitro Cellular Studies

#### 2.12.1. In Vitro Cytotoxicity Studies

The cytotoxicity of pure Qu, blank PMMs, and Qu-PMMs was performed using rat glial cells (C6) and human glioblastoma cells (U87MG) by MTT assay. The C6 and U87MG cells (1.0 × 10^4^ cells/well) were seeded in 96-well-plates (100 µL of complete DMEM media/well) for 24 h to attain 60–70% confluence. The initial stock solution of pure Qu, (100 mM) lyophilized blank PMMs, and Qu-PMMs at a 6.25 mM concentration were prepared using DMSO and PBS (7.4 pH). Later, C6 and U87MG cells were added with varying concentrations of pure Qu and Qu-PMMs (7.50, 15.62, 31.25, 62.5, 125, 250, and 500 μM). Cisplatin (100 μM) and the blank PMMs in DMEM were used as a positive control (PC) and vehicle control (VC), respectively. The cells with only media were identified as the control, and wells without cells were identified as the negative control (NC). The 96-well-plates were incubated at 5% CO_2_ with 37 ± 0.5 °C for 24, 48, and 72 h. After which, 20 µL (10 mg/mL) of MTT solution was added and incubated for 1 h to form formazone crystals. DMSO (100 µL) was added to dissolve the developed crystals after removing the media and subjected to Optical Density (OD) measurement using a microplate reader (Thermo Scientific^®^ Varioskan Flash Multimode Reader) at 570 nm [30].

The % cell viability was calculated by using the below Equation (2):% Cell viability = OD (test − blank) / OD (control − blank) × 100(2)

#### 2.12.2. Cellular Uptake Study

The intracellular accumulation of Qu and Qu-PMMs was also determined by using HPLC to compare the intracellular accumulation of PMMs with the initiation of cell death by PMMs encapsulated Qu. The C6 and U87MG cells were seeded in a 12-well plate (0.2 × 10^6^ cells/well) and incubated at 37 °C. for 36 h. The pure Qu, Qu-PMMs (equivalent to 62.5 μM Qu) were added to the cells and incubated for 3, 6, 12, and 24 h at 37 °C. Any loosely bound Qu in the incubated cells was detached by rinsing with PBS. The cells were trypsinized, and the Qu was extracted using chloroform. The chloroform was evaporated, and the samples were reconstituted in a mobile phase and analyzed using the developed HPLC method, as described in Section 2.2 [30].

#### 2.12.3. Wound Healing Migration Assay

The C6 and U87MG cells (0.5 × 106) were pre-cultured in 6-well plates for 24 h. Ensuring the cell adherence and confluence (~80%), the media was detached, and the cellular debris was removed by rinsing with PBS. A fresh DMEM media containing 1% FBS was replaced to synchronize the cell growth for 12 h. Then, a sterile 200 µL pipette tip was used to create a cell-free straight scratch/wound. The cells were washed and treated with different concentrations of Qu and Qu-PMMs (62.5, 125, and 250 µM) and incubated for 0, 24, and 48 h. The highest and lowest concentrations of Qu and Qu-PMMs were selected based on their IC50 (about 237.8 ± 2.85 μM) at 24 h of treatment. The control group was untreated, and the positive control (PC) group cells were treated with Itraconazole (5 μg/mL). After treatment, the cell migration was monitored at different time points (0, 24, and 48 h) under an inverted microscope, and ImageJ software (NIH) was used to quantify the relative cell migration [31]. The anti-migration effect of Qu and Qu-PMMs were determined and plotted as sample vs. % relative migration rate was calculated using the following Equation (3):% relative migration rate = (relative area of sample / relative area of control) × 100(3)

#### 2.12.4. Apoptosis Assay

The C6 and U87MG cells were seeded in 6-well plates at a density of 5.0 × 105 cells/well with DMEM media and incubated for 36 h to attain confluency. The cells were later treated with Qu and Qu-PMMs (62.5, 125, and 250 µM). Cisplatin (100 μM) was considered a positive control (PC), and the cells with no treatment were considered a control. The treated plates were incubated for 24 and 48 h at 37 °C. The cells were further trypsinized, collected with growth media, and centrifuged at 8000 RPM, and the cell pellets were washed with PBS and incubated for 10 min with ethidium bromide (2 μL, 100 μg/mL) and acridine orange (2 μL, 100 μg/mL) dyes. The stained cells were observed, and images were captured using a fluorescent microscope (Olympus 1X73, Tokyo, Japan) [32].

### 2.13. Statistical Analysis

Design expert software (version 13.0) was employed to determine the influence of independent variables using different statistical parameters. The obtained results are expressed in the form of mean ± SD. Statistical analysis was performed by two-way analysis of variance (ANOVA) using GraphPad Prism software, version 8.0.3, GraphPad Software, Inc., California, CA, USA.

## 3. Results

### 3.1. Formulation of Qu-PMMs

#### Model Fitting

The influence of Soluplus, E-TPGS, and poloxamer 407 on PS, PDI, and % EE are presented in Table 2. The obtained experimental data were employed to calculate the co-efficient of quadratic equation, regression equations (R2) for PS, PDI, and % EE responses. The predicted responses obtained from the equations are shown in Appendix A for PS, PDI, and % EE, respectively. The Experimental Equation resulted in regression coefficients (R^2^) for mean PS (0.8542), PDI (0.8682), and % EE (0.8491). The R2 values being closer to unity indicate a better model fitting for the obtained experimental data. In the current experiment, R2 being closer to unity demonstrates the effect of Soluplus (A), E-TPGS (B), and poloxamer 407 (C) on the PS (Y 1), PDI (Y 2), and % EE (Y 3) responses. ANOVA was conducted to determine the significance level of the coefficients. The data were evaluated independently in terms of the corresponding effect of each factor on each response. The desirability function was used for simultaneous analysis of the designed system Appendix A shows the 3D plot analysis of the desirability function.

##### Effect of Independent Variables on Responses

Particle size (PS)

The mean diameters of the prepared PMMs’ particles were in the range of 107.167 ± 1.069 to 541.567 ± 0.252. The quadratic model was the best-fitted model. ANOVA indicated that the model value (F = 4.56) and only a 2.90% chance that the F value could occur due to noise. *p* < 0.500 indicating that the model terms are significant in the present study, A, B, C, and C2 were significant factors (*p* < 0.005). Appendix A shows a 3D plot analysis of the effect of Soluplus, E-TPGS, and poloxamer 407 on the particle size of prepared PMMs.

2.Polydispersity Index (PDI)

The PDI of all the formulations ranged from 0.236 ± 0.053 to 0.803 ± 0.055. The quadratic model was the best-fitted model when PDI was considered. Soluplus and E-TPGS concentration significantly impacted PDI, showing a positive effect. As the 3D plot explains in Appendix A, the increase in Soluplus, E-TPGS, and poloxamer 407 concentration was led to an increase in PDI. ANOVA also indicated a Model F value = 5.12, which implies that the model is significant and there is only a 2.13% chance that this F value could occur due to noise, and B2 and C2 were the significant factors. 

3.Percentage entrapment efficiency (% EE)

The Qu entrapment in PMMs was found to be between 37.28 ± 3.268 and 80.37 ± 1.316%. The quadratic model was the best-fitted model (*p* < 0.05). Appendix A represents the effect of Soluplus, E-TPGS, and poloxamer 407 on % EE of prepared PMMs formulations. ANOVA suggested that the Model F value of 4.38 and F value of 3.23% could occur due to noise. Hence, factors A, C, and AB significantly impacted the % EE.

4.Verification of the model

The desirability equations for the prediction of responses were validated by reproducing the formulation as per optimum conditions (20 mg mL−1 of Soluplus, 0.5 mg mL^−1^ of E-TPGS, and 2% of poloxamer 407) predicted through an overlay plot obtained from BBD (Appendix A). The predicted response value for PS, PDI, and % EE was 109.55 nm, 0.282 mV, and 71.45%, respectively. The experimental values were in good agreement with the predicted values having PS (107.16 ± 1.069 nm), PDI (0.237 ± 0.053), and % EE (77.46 ± 1.945%). The experiment was performed in triplicate.

### 3.2. Characterization of Optimized Qu-PMMs

The single emulsion-solvent evaporation (SESE) method was utilized to prepare Qu-PMMs. Soluplus and E-TPGS were adopted to formulate the mixed micelles as they offer various advantages compared to other polymers. Figure 2III shows the optimized Qu-PMMs’ surface morphology with a spherical shape, smooth surface, and homogeneity. The PS, ZP, and blank PMMs optimized Qu-PMMs are shown in Figure 2IV(A,B). The particle size, PDI, Zeta potential and % EE of optimized Qu-PMMs and blank PMMs were found to be 107.16 ± 1.069 nm, and 77.85 ± 0.620, PDI of 0.236 ± 0.053, and 0.128 ± 0.024, Zeta potential of −1.19 ± 0.024, and −2.58 ± 0.142, % EE of 77.46 ± 1.945 and NA %, accordingly.

The FT-IR of the Pure Qu, physical mixture, blank PMMs, and Qu-PMMs are represented in Figure 2V(A–D), with their characteristic peaks. The broad peaks at 3335.96 and 3339.85 confirm the presence of the hydroxyl (OH) group in the Qu, and physical mixture, respectively. An intense peak at 1670.41 represents the conjugated ketone (C = O) group in Qu. The physical mixture and Qu peaks (Figure 2V(A,B)) represent no significant shifts in the characteristic peaks. Thus, the FTIR study concludes that the polymer mixtures are compatible with Qu.

Figure 2VI(A–D) shows the PXRD patterns of the Pure Qu, physical mixture, blank PMMs, and Qu-PMMs. Pure Qu (Figure 2VIA) exhibited the main characteristic PXRD peaks at 2θ = 10.78°, 12.48°, and 27.38°, and these 2θ were retained in the physical mixture as well (Figure 2VIB). After the SESE process, all the prominent characteristic peaks of the crystalline Qu became weak, indicating that the polymer mixtures have manipulated the crystallinity of Qu by encapsulating the drug molecule inside Soluplus and E-TPGS (Figure 2VI(C,D)).

In the in vitro drug release studies, the Qu as a pure drug released 79.24 ± 3.58% within 8 h, but the Qu-PMMs released only 38.22 ± 4.25%. About 50% of Qu was released from PMMs at 24 h (53.48 ± 8.24%), and the drug dispersed in the matrix was diffused slowly up to 72 h (92.78 ± 3.54) (Figure 2VII). The kinetic study demonstrates that Higuchi (R2 = 0.9824) is the best fitting kinetic model for optimized Qu-PMMs formulation (Table 3).

Figure 2VIII depicts the Qu-loaded PMMs stored at 4 °C and shows decreased PS on day 4 and managed to increase slightly in PS (A) from 107.16 ± 1.069 nm to 118.24 ± 1.74 nm, PDI (B) increases from 0.236 ± 0.053 to 0.243 ± 0.04 and % EE (C) decrease from 77.46 ± 1.945% to 74.31 ± 2.15% from day 0 to day 60. Similarly, Qu-loaded PMMs stored at 25 °C showed increase in PS (A) from 107.16 ± 1.069 nm to 152.30 ± 0.980 nm, PDI (B) from 0.236 ± 0.053 to 0.308 ± 0.020 and % EE (C) decrease from 77.46 ± 1.945% to 69.48 ± 1.41%. The overall PS, PDI, and % EE of Qu-PMMs stability data revealed no significant changes at both 4 and 25 °C storage conditions.

### 3.3. In Vitro Cellular Studies

#### 3.3.1. Cell Viability Assay

The MTT-based cytotoxic effect of Pure Qu and optimized Blank and Qu-PMMs were evaluated against C6 and U87MG cells. The morphological changes of the C6 and U87MG cells are represented in Figure 3I,II, respectively. With the increase in time (24, 48, and 72 h), the control C6 (spindle-shaped) and U87MG (branched and elongated) cells increased their confluency, wherein the Qu and Qu-PMMs treatment induced a round morphology to the C6 and U87MG cells. These results indicated that Qu exposure is more sensitive to glioma cells. The C6 and U87MG cells were treated with pure Qu and Qu-PMMs for dose and time-dependent responses, as shown in Figure 3III. The cytotoxicity results were compared with pure Qu, Qu-PMMs, and untreated cells designated as the control. The obtained results, the IC50 value for the C6 cells obtained for pure Qu was at 24 h (237.8 ± 2.85 μM), 48 h (137.0 ± 2.64 μM), and 72 h (71.62 ± 3.21 μM), while the IC50 value of Qu-PMMs was at 24 h (116.2 ± 4.24 μM), 48 h (52.96 ± 3.24 μM), and 72 h (32.73 ± 3.01 μM). The obtained results, the IC50 value for the U87MG cells obtained for pure Qu was at 24 h (195.5 ± 1.52 μM), 48 h (131.2 ± 1.25 μM), and 72 h (62.04 ± 3.21 μM), while the IC50 value of Qu-PMMs was at 24 h (101.2 ± 1.95 μM), 48 h (57.64 ± 3.51 μM), and 72 h (42.61 ± 3.01 μM). The IC50 values of pure Qu and Qu-PMMs suggested that the drug-loaded PMMs were 1.7-fold more than pure drugs against both glioma cells. The blank PMMs (VC) did not cause cytotoxicity and were nontoxic to the cells, revealing their biocompatibility.

#### 3.3.2. Cellular Uptake

The in vitro cellular uptake of Qu was assessed in the U87MG and C6 glioma cell lines shown in Figure 2IX(A,B), respectively. The cells were treated with Qu and Qu-PMMs, and the % cellular uptake at different time points (3, 6, 12, and 24 h) was determined by the HPLC method, as described earlier. The observed % cellular uptake of Qu and Qu-PMMs in U87MG cells was 0.59 ± 0.0091%, 3.14 ± 0.10%, 8.43 ± 0.30%, 20.77 ± 1.58% and 1.24 ± 0.052%, 9.71 ± 0.25%, 20.95 ± 2.54%, 47.19 ± 2.01% at 3, 6, 12, 24 h, respectively. Whereas the Qu and Qu-PMMs uptake in C6 cells was 0.82 ± 0.031%, 6.08 ± 0.13%, 14.61 ± 0.17%, 29.31 ± 1.02%, and 1.37 ± 0.10%, 11.61 ± 0.18%, 40.95 ± 1.65%, 62.81 ± 2.56% at 3, 6, 12, 24 h, respectively.

#### 3.3.3. Wound Healing Migration Assay

The C6 and U87MG cells were utilized to determine the ability of Qu PD and Qu-PMMs to decrease the % relative cell migration (complete closure of wounded gap) was measured using wound healing migration assay. The cells were cultured and treated with different doses (0, 62.5, 125, and 250 μM) of Qu PD and Qu-PMMs for different time intervals (0, 24, and 48 h). A qualitative result of the time and dose-dependent treatment on the C6 and U87MG cells showed the increased width of the wounded gap (yellowed line) compared to the control (untreated) cells, represented in Figure 4A(C6),B(U87MG).

A quantitative report resulted a notable difference in the % relative cell migration of U87MG cells treated with Qu PD 48.01 ± 6.24%, 45.32 ± 7.21% and 29.69 ± 2.40% (24 h incubation) and 71.67 ± 3.95%, 69.67 ± 4.84% and 37.96 ± 6.45% (48 h incubation) compared with Qu-PMMs 37.90 ± 4.20%, 32.54 ± 1.08% and 12.20 ± 2.14% (24 h incubation) and 52.63 ± 2.14%, 37.78 ± 3.25%, and 18.50 ± 3.21% (48 h incubation) with a concentration of 62.5, 125, and 250 μM, respectively (Figure 4C). The migration ability of control and PC on U87MG cells were shown as 58.57 ± 12.01% (24 h), 91.95 ± 6.58% (48 h) and 46.65 ± 1.21% (24 h), 55.76 ± 2.58% (48 h), respectively. A quantitative report resulted a notable difference in the % relative cell migration of the C6 cells treated with Qu PD 44.76 ± 3.40%, 37.80 ± 2.51% and 9.80 ± 2.14% (24 h incubation) and 60.72 ± 5.25%, 57.22 ± 8.05% and 10.33 ± 2.53% (48 h incubation) compared with Qu-PMMs (12.66 ± 1.21%, 10.31 ± 1.78% and 2.57 ± 0.68% (24 h incubation) and 24.93 ± 3.47%, 14.00 ± 5.25% and 2.71 ± 0.95% (48 h incubation) with concentration of 62.5, 125 and 250 μM, respectively. The migration ability of the control and PC on C6 were shown to be 55.76 ± 8.51% (24 h), 84.13 ± 6.58% (48 h), and 49.91 ± 2.85% (24 h), 56.49 ± 3.84% (48 h), respectively (Figure 4D). The migration ability of the control and PC on C6 were shown as 10.32 ± 0.85% (24 h), 12.30 ± 1.02% (48 h) and 45.18 ± 3.56% (24 h), 53.03 ± 4.21% (48 h), respectively.

These results indicated that the Qu-PMMs treatment on both the C6 (**** *p* ˂ 0.0001) and U87MG (*** *p* ˂ 0.001) cells were significantly higher (2 to 4-fold) than Qu PD. The cell migration results indicated that the % relative migration of the control was more when compared to the Qu treatment.

#### 3.3.4. Apoptosis Assay

Apoptosis was analyzed and quantified in the C6 and U87MG cells using acridine orange/ethidium bromide (AO/EtBr) under a fluorescence microscope. We estimated the cell death by assessing the AO/EtBt staining after 24 and 48 h of incubation with different doses (0, 62.5, 125, and 250 μM) of Qu PD and Qu-PMMs. A qualitative result of Qu treatment on the C6 and U87MG cells showed increased cell death (red colored cells) compared to the control (red colored cells), picturized in Figure 5A(U87MG), B(C6). A quantitative report resulted a notable difference in the % cell death of the U87MG cells treated with Qu PD 31.36 ± 1.12%, 37.71 ± 2.51% and 40.40 ± 4.85% (24 h incubation) and 40.22 ± 2.54%, 51.82 ± 2.21% and 52.39 ± 2.36% (48 h incubation) compared with Qu-PMMs 41.97 ± 2.45%, 43.55 ± 1.65% and 47.72 ± 2.45% (24 h incubation) and 48.02 ± 1.34%, 70.82 ± 3.54% and 75.25 ± 2.15% (48 h incubation) with the concentrations of 62.5, 125, and 250 μM, respectively. These results indicated that the Qu-PMMs treatment on the C6 and U87MG (*** *p* ˂ 0.001) cells was significantly higher (1.7-fold) than Qu PD. The migration ability of the control and PC on the U87MG cells were shown to be 12.13 ± 0.25% (24 h), 19.59 ± 1.20% (48 h) and 25.58 ± 1.25% (24 h), 43.50 ± 1.08% (48 h), respectively (Figure 5C). A quantitative report resulted a notable difference In the % cell death of C6 cells treated with Qu PD 12.01 ± 1.58%, 28.76 ± 1.14% and 36.40 ± 2.54% (24 h incubation) and 26.31 ± 2.95%, 47.20 ± 1.02% and 52.14 ± 3.13% (48 h incubation) compared with Qu-PMMs (18.69 ± 2.05%, 34.71 ± 3.96% and 55.51 ± 2.95% (24 h incubation) and 50.55 ± 6.58%, 67.65 ± 4.18% and 86.81 ± 4.18% (48 h incubation) with the concentrations of 62.5, 125, and 250 μM, respectively.

## 4. Discussion

The present study w aimed to develop and evaluate Qu-loaded polymeric mixed micelles by single emulsion solvent evaporation method. Amphiphilic polymers with lower critical micellar concentration (CMC) are preferred for micellar formulation as CMC plays a crucial role in in vitro, in vivo, and thermodynamic stability. Moreover, it is essential to increase the solubilization capacity of polymeric micelles (PMs) [33]. Various findings suggested that the addition of E-TPGS and poloxamer reduced the CMC of Soluplus [19,34,35]. In the present study, the primary goal was to identify and optimize the quantity of the Soluplus, E-TPGS, and Poloxamer 407 combination in Qu-PMMs by applying DoE and evaluating the delivery and in vitro performance on glioma cells.

In the optimization, the mean PS and PDI were increased as Soluplus decreased, and E-TPGS and poloxamer 407 concentrations were increased as the incorporation or aggregation of excess polymer molecules overlapped on the already formed micelles. Furthermore, as the polymer concentration increases, the % EE is enhanced. Hence, it was concluded that the preparation of PMMs with uniform particles, optimum polymer, and stabilizer concentration need to be selected cautiously.

The particle size of <200 nm can passively target the cancer site by EPR (enhanced permeability and retention) effect, increased distribution of drugs in the body, and reduced side effects to normal cells [35]. The less PDI depicts uniform-sized and monodispersed PMMs, which confirms the successful fabrication of mixed micelles. The negative zeta potential (ZP) results of blank and Qu-PMMs were due to the mixed hydrophilic polymers containing -OH functional groups [36]. The % EE ensures the ability of the chosen nanocarriers to encapsulate and deliver the effective dose of drugs to the intended site.

The FTIR spectra of the physical mixture retained a peculiar corresponding peak of Qu pure drug (Figure 2V(A,B)). However, the broad peak at 3466.49 and 3466.16 (intermolecular hydrogen bonding) of the blank PMMs and Qu-PMMs (Figure 2V(A,B)) may enhance the solubility, and this suggests that the Qu drug encapsulated inside the carrier [37]. Further, the P-XRD results imply Qu’s successful incorporation within the PMMs in an amorphous form. Figure 2 VI(C,D) shows a similar profile to the blank and Qu-PMMs, revealing that the outer shell consisted of Soluplus and E-TPGS with a sheath of Poloxamer 407 over the micelles. Hence, the results confirmed micelles’ core–shell structure incorporating the drug inside the core within the polymeric shell.

A comparatively higher drug release of pure Qu (Figure 2VII) within three hours than the Qu- PMMs was attributed to a minor Qu release (sustained-release) due to the encapsulation of the Qu drug in Soluplus. The Qu release data follow the Higuchi model shown in Appendix A. Furthermore, the PS, PDI, and % EE of the optimized Qu-PMMs had no significant change after 60 days of storage at 4 and 25 ⁰C, which could be possible due to the low CMC value of combined Soluplus, E-TPGS, and poloxamer 407 [34,38]. The high stability of the optimized formulation suggests the ability to maintain integrity even upon dilution in the body. The in vitro characterization study resembled previous investigations [22,33,35].

The in vitro % cell viability of C6 and U87MG cells in the treatment of Qu and Qu-PMMs was evaluated by MTT assay. Qu depicted the necrotic morphological changes in glioma cells and decreased the % cell viability in a dose and time-dependent manner. The toxicity results of the blank PMMs on C6 and U87MG cells were low, around 2% during 72 h of exposure time. Additionally, the Qu-PMMs treatment had a significant effect on inhibiting the glioma cells compared to pure Qu. Considering these results, it may conclude that the encapsulation of Qu into micelles has decreased its % cell viability by around 1.7-fold compared to pure Qu on glioma cells, incredibly long incubation period, which is probably due to the high stability sustained release pattern of the optimized PMMs. It can be correlated with the previous study showing the results of the % cell viability on C6 cells of Qu encapsulated polymeric nanoparticles also had the lowest IC_50_ than pure Qu [10].

The % cellular uptake was determined by using the HPLC method as mentioned in Section 2.2 to treat Qu and Qu-PMMs on C6 and U87MG glioma cells. The 62.5 µM dose was selected due to less anti-proliferating activity. These results showed that the Qu-PMMs were 2.12 and 2.41-fold increased than Qu pure drug in the C6 and U87MG cells. The Qu-PMMs depicted higher penetration, targeting, and drug diffusion inside cells due to decreased particle size and stability of nanocarriers. The maximum uptake (62.85 ± 2.56%) of Qu-PMMs was observed within 24 h of the study, wherein the highest toxicity was exhibited with a long period of incubation (48 and 72 h), which correlates with drug release study. The % drug uptake increased as time increased in both glioma cells (Figure 2IX(A,B)).

The time and dose-dependent anti-migration and apoptosis were found in both glioma cells. The reduced % relative cell migration of C6 and U87MG cells was observed after the Qu and Qu-PMMs treatment. The lower concentration of Qu had no cell migration with 24 h of incubation, wherein the same concentration had significant inhibition in both the cells. The Qu-PMMs had a markable impact on reducing cell migration by increasing exposure time, also previously documented [1]. Apoptosis assay results also indicated similar findings with Qu-PMMs. The decreased cell migration and the higher apoptotic ratio of the Qu-treated cells support the higher cellular uptake of Qu-PMMs and the decreased % cell viability of both C6 and U87MG glioma cell lines [10].

These in vitro physicochemical and cell line data confirmed that the performance of Qu-PMMs is higher than pure Qu. This study has fulfilled our purpose by enhancing the stability with the combination of polymers and also exhibited the maximum effect in glioma cells compared to the control and pure Qu. Hence, in the future, work can be extended in a similar direction to develop a polymeric hybrid lipid nanoparticle to enhance the blood-brain barrier penetration for successful application in the treatment of Glioma.

## 5. Summary and Conclusions

In the current study, polymeric mixed micelles (PMMs) were prepared using the single emulsion-solvent evaporation method and optimized using Box-Behnken Design. The influence of the independent variables on the PS, PDI, and % EE of Qu-PMMs was studied successfully. The SEM, FTIR, and P-XRD resulted in the optimized Qu-PMMs being spherical, having maximum encapsulation inside the carrier, and stable over 60 days. In vitro drug release studies showed that the Qu was released in a sustained manner from PMMs. Furthermore, the in vitro cell-based assays have confirmed the improved safety and efficacy of Qu-PMMs. Based on the constructive results from various characterizations, we conclude that the PMMs can be utilized as effective drug carriers to deliver lipophilic drugs.

## Figures and Tables

**Figure 1 pharmaceutics-14-01643-f001:**
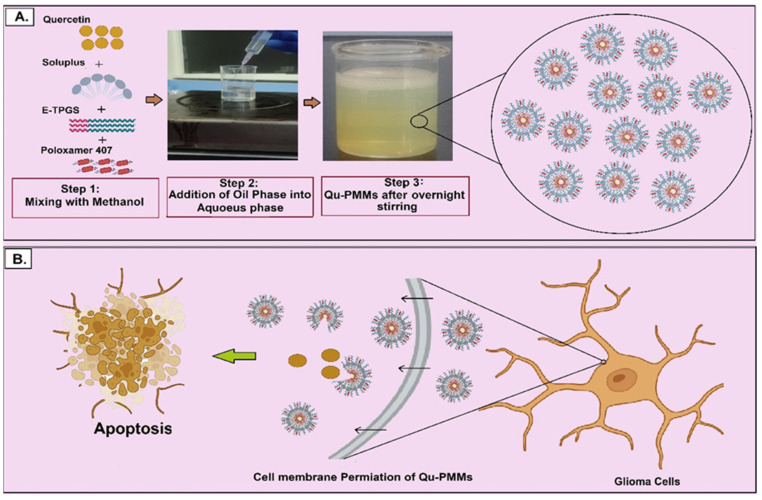
Schematic representation of the preparation and mode of action of polymeric mixed micelles (PMMs); (**A**) Preparation of PMMs using single emulsion solvent evaporation method; (**B**) Graphical illustration of in vitro glioma cell death on the treatment of Qu-PMMs.

**Figure 2 pharmaceutics-14-01643-f002:**
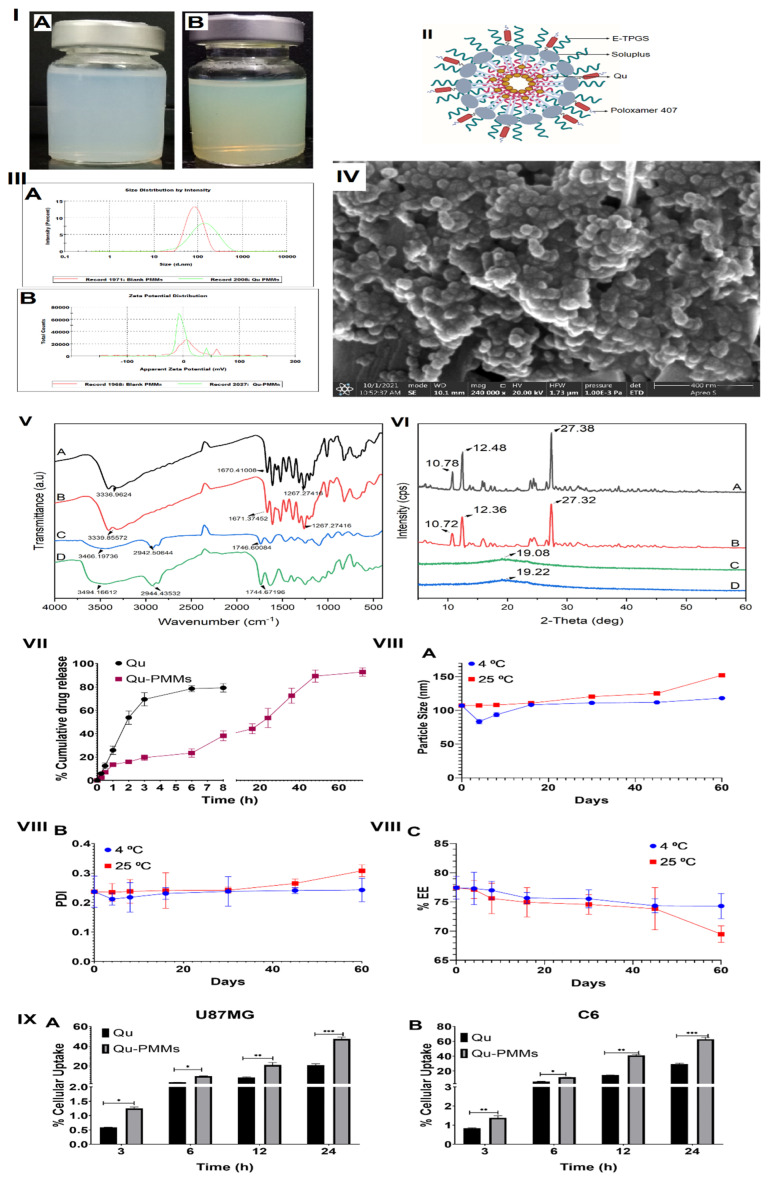
In vitro evaluations of optimized Qu-PMMs; (**I**) Physical appearance of (**A**) blank PMMs and (**B**) Qu-PMMs optimized formulation; (**II**) Schematic illustration of Qu-PMMs; (**III**) Particle size (**A**) and zeta potential (**B**) analysis of blank PMMs and Qu-PMMs; (**IV**) Scanning electron microscopy (SEM) morphology evaluation of Qu-PMMs; (**V**) Drug release profile of Qu-PMMs; (**VI**) Stability analysis of (**A**) PS, (**B**) PDI, and (**C**) % EE; IX. In vitro % cellular uptake of (**A**) U87MG and (**B**) C6 cells on Qu and Qu-PMMs treatment. Data are represented in means ± Standard deviation (*n* = 3). * Indicates *p* ˂ 0.05, ** represents *p* ˂ 0.01 and *** represents *p* ˂ 0.001 (Qu PD vs. Qu-PMMs).

**Figure 3 pharmaceutics-14-01643-f003:**
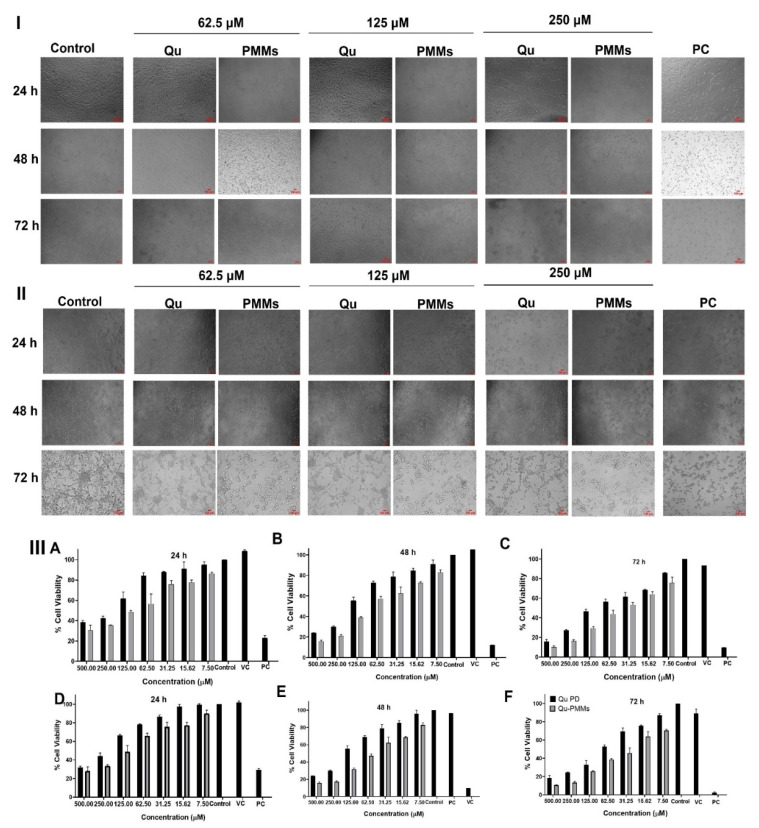
Cell Viability Assay of Qu and Qu-PMMs treatment on C6 and U87MG Cells; Pictorial representation of morphological changes of C6 (**I**) and U87MG (**II**) cells on the treatment of Qu and Qu-PMMs; (**III**) % Cell viability of C6 and U87MG cells treated Qu and Qu-PMMs time ((**A**,**D**) (24 h), (**B**,**E**) (48 h) and (**C**,**F**) (72 h)) and dose (7.50, 15.62, 31.25, 62.5, 125, 250, and 500 μM) dependent studies by using MTT assays. Data are represented in means ± Standard deviation (*n* = 3). Scale bar (**A**,**B**) (10×) 100 µm.

**Figure 4 pharmaceutics-14-01643-f004:**
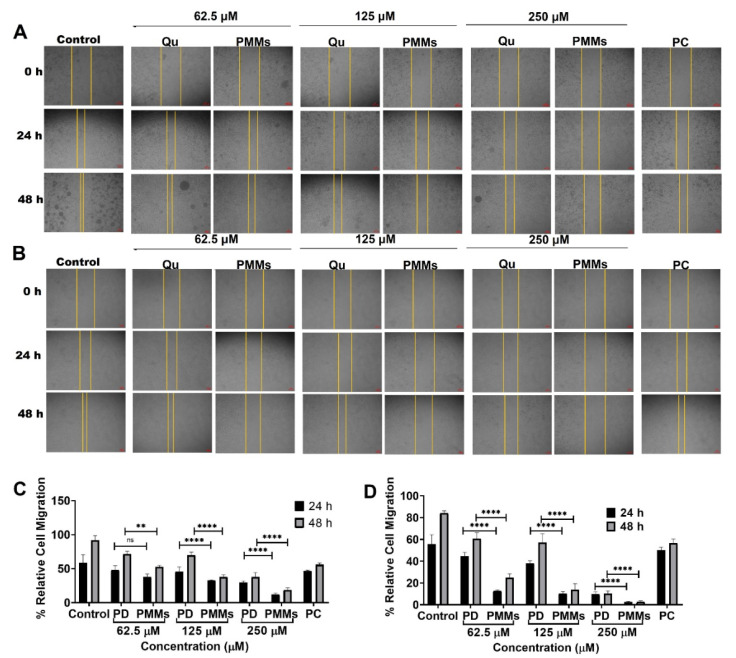
C6 and U87MG cells migration assay. (**A**) Pictorial representation of morphological changes of U87MG (**A**) and C6 (**B**) cells on the treatment of Qu and Qu-PMMs; The graphical representation of % relative cell migration of U87MG (**C**) and C6 (**D**) cells treated Qu and Qu-PMMs time (0 h, 24 h, and 48 h) and dose (62.5, 125, and 250 μM) dependent migration study by wound healing assays. Data are represented in means ± Standard deviation (*n* = 3). ns Indicates non-significant, ** represents *p* ˂ 0.01, and **** denotes *p* ˂ 0.0001 (Qu PD vs. Qu-PMMs). Scale bar (**A**,**B**) (4×) 100 µm.

**Figure 5 pharmaceutics-14-01643-f005:**
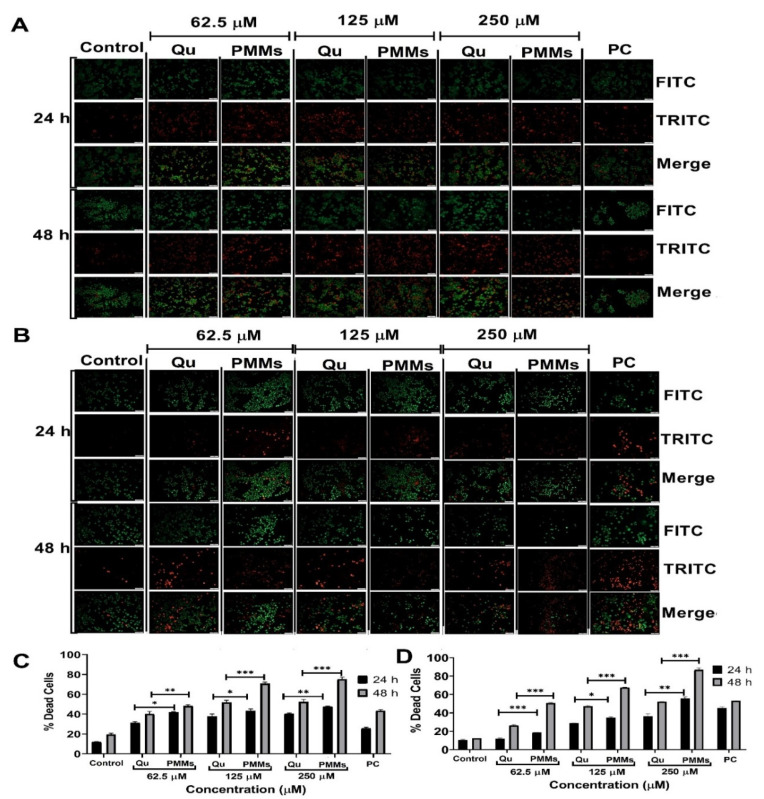
C6 and U87MG cells Apoptosis Assay. A. Pictorial representation of morphological changes of U87MG (**A**) and C6 (**B**) cells on the treatment of Qu and Qu-PMMs; The graphical representation of % apoptosis of U87MG (**C**) and C6 (**D**) cells treated Qu and Qu-PMMs time (24 h and 48 h) and dose (62.5, 125, and 250 μM) dependent migration study by wound healing assays. Data are represented in means ± Standard deviation (*n* = 3). * Indicates *p* ˂ 0.05, ** represents *p* ˂ 0.01 and *** represents *p* ˂ 0.001 (Qu PD vs. Qu-PMMs). Scale bar A (10×) 50 µm and B (20×) 100 µm.

**Table 1 pharmaceutics-14-01643-t001:** Independent and dependent variables in Box Behnken design for preparation and optimization of Qu-PMMs.

Factors	Levels		
Independent Variable	Low (−1)	Centre (0)	High (+1)
A = Soluplus (mg mL^−1^)	10	15	20
B = E-TPGS (mg mL^−1^)	0.1	0.5	1.0
C = Poloxamer 407 (% *w/v*)	0.1	0.2	0.3
Dependent variable	Goals		
Y1 = PS (nm)	Decrease		
Y2 = PDI	Decrease		
Y3 = EE (%)	Increase		

**Table 2 pharmaceutics-14-01643-t002:** BBB designs factors and Observed responses for Qu-PMMs.

Run	A	B	C	Y1	Y2	Y3
1	20	0.5	0.3	187.700 ± 8.654	0.398 ± 0.057	79.800 ± 0.721
2	20	0.1	0.2	107.167 ± 1.069	0.236 ± 0.053	77.460 ± 1.945
3	15	0.5	0.2	228.400 ± 8.190	0.296 ± 0.055	57.000 ± 1.952
4	20	1.0	0.2	192.320 ± 5.155	0.269 ± 0.014	59.920 ± 2.316
5	15	1.0	0.3	385.030 ± 6.615	0.330 ± 0.025	80.370 ± 1.316
6	15	0.5	0.2	231.730 ± 5.139	0.283 ± 0.044	55.610 ± 1.561
7	10	1.0	0.2	184.100 ± 3.820	0.264 ± 0.020	60.620 ± 1.660
8	15	0.1	0.1	541.560 ± 0.252	0.803 ± 0.055	55.550 ± 0.547
9	20	0.5	0.1	158.730 ± 5.519	0.345 ± 0.019	43.670 ± 0.846
10	15	0.5	0.2	228.400 ± 8.190	0.289 ± 0.046	76.330 ± 1.030
11	15	0.1	0.3	194.500 ± 1.706	0.393 ± 0.020	64.260 ± 0.981
12	15	1.0	0.1	299.430 ± 5.123	0.648 ± 0.021	72.310 ± 0.993
13	10	0.5	0.1	294.500 ± 1.706	0.393 ± 0.020	41.980 ± 1.547
14	10	0.1	0.2	321.960 ± 0.252	0.364 ± 0.020	37.280 ± 3.268
15	10	0.5	0.3	441.560 ± 9.075	0.397 ± 0.043	43.670 ± 0.846
16	15	0.5	0.2	225.060 ± 10.696	0.273 ± 0.011	58.000 ± 2.157
17	15	0.5	0.2	226.730 ± 9.078	0.281 ± 0.020	57.330 ± 2.521

A = Soluplus (mg mL^−1^), B = E-TPGS (mg mL^−1^), C = poloxamer 407 (% *w/v*), Y1 = Particle size (nm), Y2 = PDI and Y3 = Entrapment Efficiency (%). Data are represented in means ± Standard deviation (*n* = 3).

**Table 3 pharmaceutics-14-01643-t003:** Release kinetics of Qu-PMMS.

Zero Order		First Order		Higuchi		Korsmeyer-Pappas	
R^2^	K_0_ (h^−1^)	R^2^	K_1_ (h^−1^)	R^2^	K_H_ (h^−1/2^)	R^2^	K_KP_ (h^−n^)
0.8939	0.5175	0.8757	1.257	0.9824	7.409	0.9429	0.8892

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
