# Peer review of "Anti-Proliferative Potential of Quercetin Loaded Polymeric Mixed Micelles on Rat C6 and Human U87MG Glioma Cells"

_pharmaceutics, 2022, doi:10.3390/pharmaceutics14081643_

Round 1

Reviewer 1 Report

The authors synthesized Qu-loaded polymeric mixed micelles. The higher loading drug efficiency, improved drug releasing capability and higher cytotoxicity demonstrates the enhanced therapeutic effect of Qu-PMMs and can be considered an effective therapeutic strategy to treat Glioma.

1. From the Figure 4C the difference between Qu and PMMs are so subtle in the figure but p value is still smaller than 0.05 in 62.5 uM (both 24 h and 48 h cases) and 125 uM (24 h case)? Could the authors provide more evidences to identify the significant difference between Qu and PMMs in the migration assay?

2. It would be great that the authors can elaborate more the rationale on utilizing wound healing migration assay to measure drug releasing capability between Qu and PMMs. I didn't see the robustness of PMMs over Qu on drug releasing in Figure4c. Could the authors explain the root cause of different behavior of both Qu and PMMs in U87MG cells and C6 cells?

Reviewer 2 Report

The current manuscript entitled "Anti-proliferative potential of Quercetin loaded polymeric mixed micelles on rat C6 and human U87MG Glioma Cells" shows the greater anticancer activity of quercetin as a polymeric mixed micelles formulation. The authors have described the preparation and characterization of the formulation in details and also investigated the activity of quercetin and its polymeric mixed micelles formulation against  rat C6 and human U87MG Glioma cells. The authors need to revise the manuscript in order to make it acceptable to Pharmaceutics.

Introduction, lines 47-48: Format the reference according to journal style.

Lines 49-50: Provide appropriate references in order to support your statement.

Line 161: Is it 01 ml or 10 ml?

Methodology: Provide more details in SEM methodology.

Apoptosis assay: Line 244: change 105 to 105

Figure 3 was mentioned as Figure 5 in legend. Correct it.  The authors need to provide better quality of Figures 3 I and 3 II. It is very difficult to draw any conclusion from these figures.

Figures 4A and B. Again the quality of the figures is very poor.

Statistical analysis: Did authors use any post-test for statistical analysis?

Round 2

Reviewer 2 Report

Authors have satisfactorily responded to my comments.